# Rain Attenuations Based on Drop Size Distribution (DSD) Model and Empirical Model at Low THz Frequencies

**Yongho Kim** [1], **Jongho Kim** [2] , **Jinhyung Oh** [2], **Youngkeun Yoon** [2], **Sangwook Park** [3,*] and **Jaegon Lee** [1,*]

[1] Department of Electronic and Software Engineering, Kyungnam University, Changwon 51767, Republic of Korea
[2] Electronics and Telecommunications Research Institute, Daejeon 34129, Republic of Korea; ykyoon@etri.re.kr (Y.Y.)
[3] Department of Electronic Engineering, Soonchunhyang University, Asan 31538, Republic of Korea
[*] Correspondence: wave@sch.ac.kr (S.P.); jaegonlee@kyungnam.ac.kr (J.L.)

**Abstract:** Rain attenuation based on the drop size distribution (DSD) with different rainfall rates ($R$) at low THz frequencies is investigated in this paper. The rain attenuation is calculated using the DSD measured for one year and the extinction cross-section (ECS) by the Mie scattering theory. Moreover, the obtained specific rain attenuation is verified by the empirical model using the measurement system consisting of a transmitter, a receiver, and weather measurement units. We measured the received power against the uniform transmitted power at 240, 270, and 300 GHz on the rooftop of the National Radio Research Agency (RRA) in Korea during the same period as the DSD measurement period. After curve fitting by regression analysis, we compared both rain attenuations obtained in two methods with the recommendation International Telecommunication Union Radiocommunication Sector (ITU-R) P.838-3. The root mean square errors (RMSEs) of the DSD model are 2.8977, 2.8646, and 2.8331 at 240, 270, and 300 GHz, respectively. The calculated result using the Mie scattering and the measured DSD methods shows the best fit to the data of the ITU-R recommendation for a rainfall rate of up to 5 mm/h. On the other hand, the empirical results using the T/Rx antenna system are slightly higher compared to the data of the ITU-R recommendation. As the rainfall rate increases, the difference between our results and ITU-R recommendation increases. This study will be useful for predicting rain attenuation for terrestrial wireless links operating at low THz frequencies.

**Keywords:** rain attenuation; drop size distribution (DSD) model; empirical model; Mie scattering; low THz frequency





## 1. Introduction

In order to establish a terrestrial wireless communication network, it is necessary to anticipate attenuation caused by atmospheric gases and particulates such as rain and snow. The atmospheric gases and particulates mainly give rise to the absorption and scattering of electromagnetic (EM) waves, respectively. In particular, the rain attenuation models in terrestrial links are classified based on the formulation of the rain attenuation model in [1]. These include the empirical [2–4], physical [5,6], statistical [7], fade slope [8,9], and optimization-based models [10,11]. The empirical model is based on experimental data observations rather than input–output relationships that can be mathematically described. The physical model is based on some of the similarities between the rain attenuation model's formulation and the physical structure of rain events. The statistical model, including the ITU-R model, is based on statistical weather and infrastructural data analysis, and the final model is built as a result of regression analysis in most cases. In the fade slope model, the slope of attenuation from the rain attenuation versus time data was developed with a particular experimental setup. In the optimization-based model, the input parameters of some of the other factors that affect rain attenuation are developed through an optimization process. Since there is an increasing demand for the use of

EM spectrum from 0.1 to 1 THz for ultrahigh-speed wireless communication and indoor applications such as remote sensing, various studies on attenuation for rain in the low THz band have been published in recent years [12–23]. As the frequency increases within the mm wave and low THz bands, the raindrop size distribution (DSD) should be considered due to the shorter wavelength compared with the size of the raindrop. The prediction model of rain attenuation can be broadly classified into two categories: the empirical model, which is based on measurement through an antenna system, and the physical model using DSD and scattering in particulates. However, most research predicts rain attenuation using the theoretical DSD models and measurement through wireless links. In [12], the specific rain attenuation is calculated using the DSD obtained by the measurement system at 1, 100, and 1000 GHz, but the rain attenuation value is not compared with a measured attenuation. In [15], the DSD value for the specific rain attenuation is obtained by the analytic model using a rainfall rate, and the attenuation value is not verified with a measured attenuation. Ref. [19] compares the specific rain attenuations using a physical model and empirical model, but the DSD value for the specific rain attenuation is obtained with the analytic model using a rainfall rate. The rain attenuation based on the physical model is calculated using the measured DSD and is verified using an empirical model [21]. However, the comparison is not performed at the same time.

In this paper, the rain attenuation based on the physical model is calculated with different rainfall rates (*R*) at 240, 270, and 300 GHz and is verified with the empirical model. Then, both rain attenuations obtained in the two methods are compared with the recommendation ITU-R P.838-3 [24]. The recommendation ITU-R P.838-3 is a curve fit with a double physical and empirical nature, to maximize the prediction accuracy on the DBSG5 database of ITU-R Study Group 3 for attenuation long-term statistics. The physical model is achieved by measured DSD and Mie scattering for spherical drops [25]. The DSD was measured for one year (from March 2021 to February 2022) using a laser-optical disdrometer (OTT Parsivel$^2$) that classifies raindrops into 1 of 32 separate size and velocity classes. Also, the extinction cross-section (ECS), which is basically the sum of the absorption and scattering cross-sections, is needed for the physical model. Then, the ECS is calculated using Mie scattering because it is more appropriate to apply Mie scattering rather than Rayleigh scattering, considering the frequency and size of the raindrop. Additionally, the empirical model is created by the measurement system consisting of a transmitter, a receiver, and weather measurement units for the place where the DSD was measured. The weather measurement unit collects rainfall rates with atmospheric temperature, humidity, and snowfall to distinguish rain attenuation from atmospheric gas attenuation. In this paper, the rain attenuation calculated by the ECS and the measured DSD is described in Section 2. Section 3 presents the measurement system and empirical model of rain attenuation eliminating atmospheric gas attenuation. In Section 4, the rain attenuations obtained by the two methods are compared and analyzed with ITU-R recommendations about specific attenuation models for rain for use in prediction methods. Finally, the conclusions are presented in Section 5.

## 2. Drop Size Distribution (DSD) Model for Rain Attenuation

The DSD was calculated using a laser-optical disdrometer (OTT Parsivel$^2$) on the rooftop of the National Radio Research Agency (RRA) in southwestern Korea from March 2021 to February 2022. OTT parsivel$^2$, which is a laser-based optical system for complete and reliable measurement of all types of precipitation, is composed of two sensor heads with splash protection unit grid, tunnel housing with 30 mm wide and 180 mm long light strip, base holder with integrated electronics and 8-pin panel jack for connecting the supply voltage and electrical parts. The disdrometer classifies raindrops into one of 32 separate size

and velocity classes. The scale of this classification is smaller for small and slow particles than for large and fast particles. The DSD can be obtained by Equation (1) [26].

$$N(D_i) = \sum_j \frac{n(D_i, v_j)}{v_j} \cdot \frac{1}{S \cdot \Delta t \cdot dD_i} \tag{1}$$

where $v_j$ and $n(D_i, v_j)$ are the median of raindrop velocity corresponding to class $j$ and the number of raindrops with size of $D_i$ and velocity of $v_j$, respectively. Also, $S$ is the area of measurement, and $\Delta t$ is the measurement time. $D_i$ is the median of raindrop size corresponding to class $i$, and $dD_i$ is the span of $i$ class that is decided by the specification of measurement equipment. To confirm the validity of our DSD measured by OTT Parsivel$^2$, the measured DSD is plotted together with the previous proposed MP [14], gamma [27], and Weibull [28] DSD models for various rainfall rates in Figure 1. The number of drops decreases markedly when the drop size diameter increases for all rainfall rates.

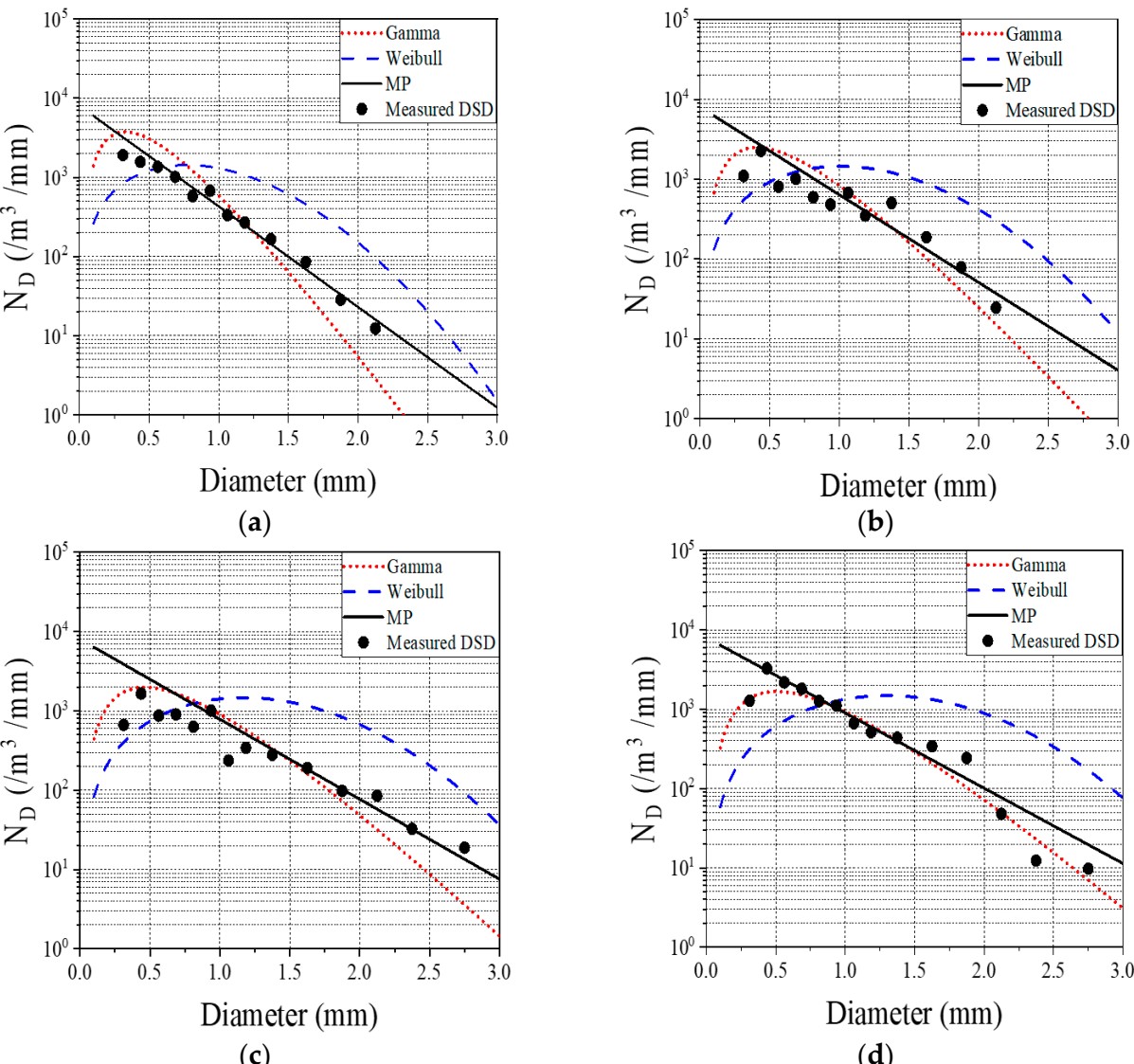

**Figure 1.** Measured DSD with gamma, Weibull, and MP DSD models for (**a**) 5 mm/h, (**b**) 10 mm/h, (**c**) 15 mm/h, and (**d**) 20 mm/h.

The extinction cross-section (ECS) is calculated by Mie scattering for spherical drops at 240, 270, and 300 GHz [24]. The ECS is thereby determined by

$$\sigma_{ext}(D_i) = \frac{4\pi}{k^2} ReS(0) \tag{2}$$

where $S(0)$ is the forward scattering coefficient, and $k$ is the wavenumber. The forward scattering coefficient can be obtained by

$$S(0) = \frac{1}{2}\sum_{n=1}^{n=\infty} (2n+1)(a_n + b_n) \tag{3}$$

where $a_n$ and $b_n$ are extinction efficiency and scattering efficiency, respectively. Although $n$ is defined as infinity in Equation (3), it can be replaced with a real value of $kD + 4(kD)^{1/3} + 2$.

$$a_n(m,\alpha) = \frac{m^2 j_n(m\alpha)[\alpha j_n(\alpha)]' - j_n(\alpha)[m\alpha j_n(m\alpha)]'}{m^2 j_n(m\alpha)\left[\alpha h_n^{(1)}(\alpha)\right]' - h_n^{(1)}(\alpha)[m\alpha j_n(m\alpha)]'} \tag{4}$$

$$b_n(m,\alpha) = \frac{j_n(\alpha)[m\alpha j_n(m\alpha)]' - j_n(m\alpha)[\alpha j_n(\alpha)]'}{h_n^{(1)}(\alpha)[m\alpha j_n(m\alpha)]' - j_n(m\alpha)\left[\alpha h_n^{(1)}(\alpha)\right]'} \tag{5}$$

where $j_n(\alpha)$ and $h_n^{(1)}(\alpha)$ are the spherical Bessel function of the first kind and the spherical Hankel function of the first kind, respectively. $m$ is a complex refractive index of raindrops at 20 °C. The specific rain attenuation using the DSD [24] is calculated by

$$\gamma = \frac{10}{\ln(10)}\sum_i \sigma_{ext}(D_i){\cdot}N(D_i){\cdot}dD_i \tag{6}$$

Figure 2 shows the specific rain attenuation using ECS and measured DSD at 240, 270, and 300 GHz. To compare the specific rain attenuation ($\gamma_R$) between our DSD model and ITU-R P.838.3 model in Section 4, we have performed a curve fit using regression analysis. It is assumed that the shape of the curve-fitting graph is Equation (7) to match the data using measured DSD effectively, and to have a similar form to the recommendation ITU-R P.838-3.

$$\gamma_R = kR^\alpha \tag{7}$$

where $k$ and $\alpha$ are the coefficients for frequency and polarization. The skewness in the distribution of residuals is also examined, as shown in Figure 3, in order to test the hypothesis that the residuals, as determined with this test variable, were distributed normally. The residual is the difference between the common logarithm value for the specific rain attenuation calculated through DSD and the common logarithm value for the curve fitting for each frequency. To analyze the distribution of the residuals, the histogram bin is set to 0.02. The red curve is a curve that converts the probability density to relative probability by multiplying the normal distribution by the histogram bin 0.02 to compare the normal distribution obtained through the mean and standard deviation of the residual with the histogram in the probability. Figure 4 shows the calculated standard deviation of the residual against frequencies. As shown in Figure 4, the dispersion of rain-specific attenuation around its fitting values is high in all the considered frequencies. It increases up to 200 GHz and then saturates. Table 1 shows the coefficient values and root mean square errors (RMSEs) based on the measured DSD model at 240, 270, and 300 GHz. The $k$ value and $\alpha$ value are smaller and larger than those of recommendation ITU-R P.838-3, respectively.

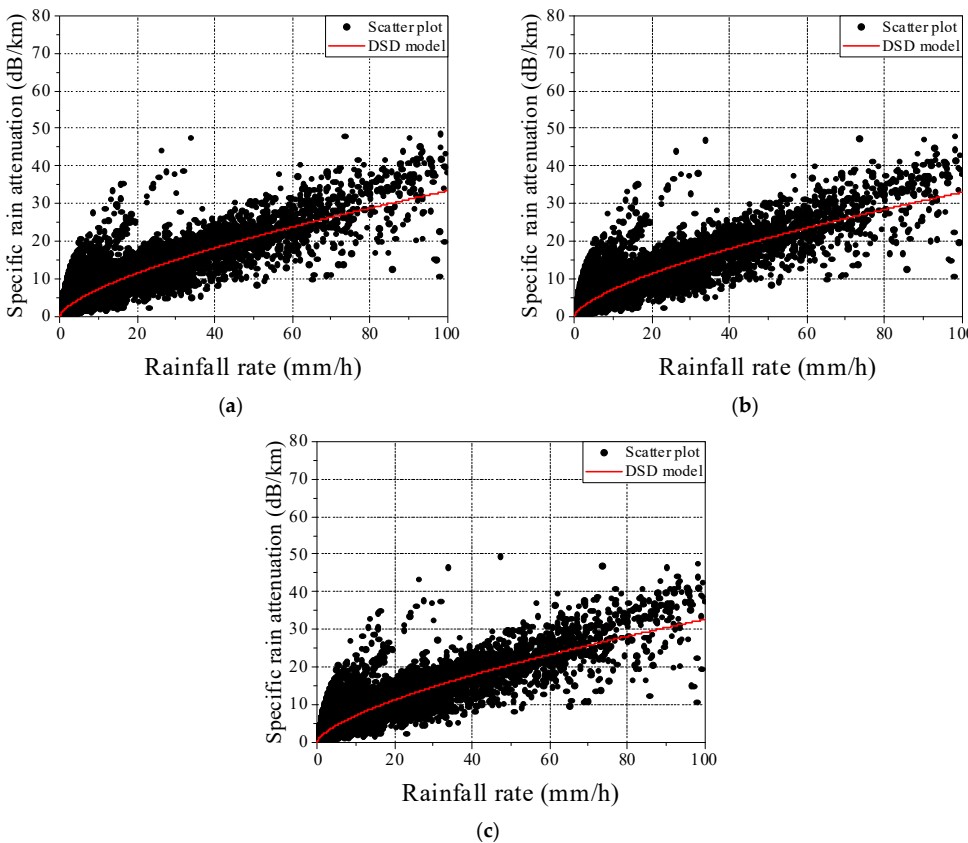

**Figure 2.** Specific rain attenuation using ECS and measured DSD: (**a**) 240 GHz, (**b**) 270 GHz, and (**c**) 300 GHz.

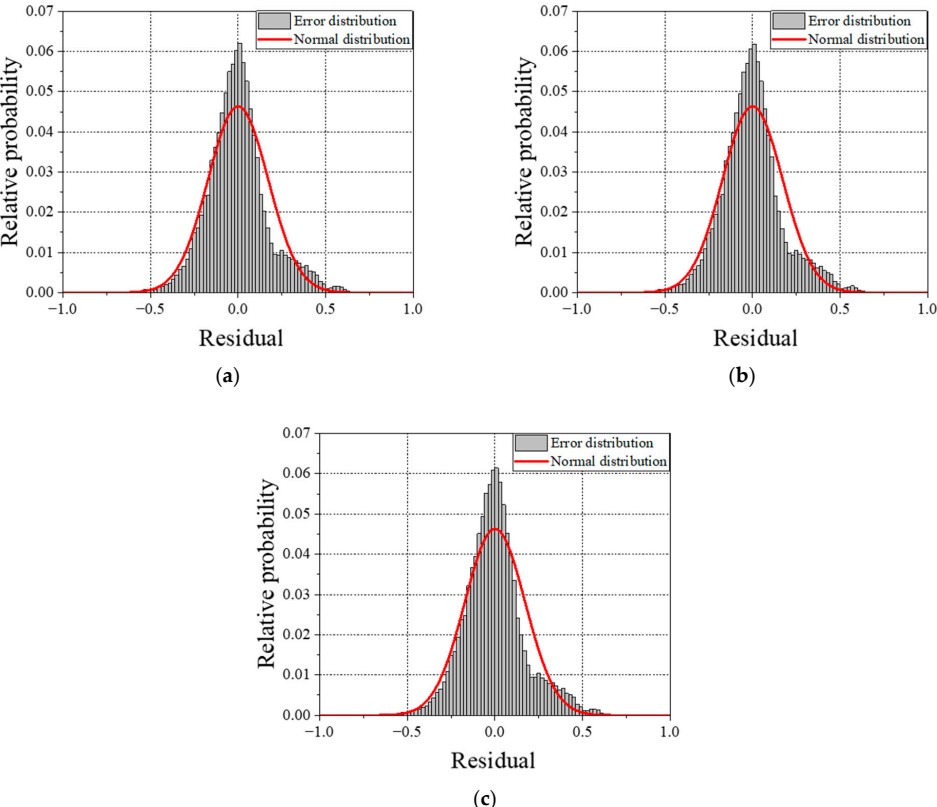

**Figure 3.** Relative probability distribution of the residual: (**a**) 240 GHz, (**b**) 270 GHz, and (**c**) 300 GHz.

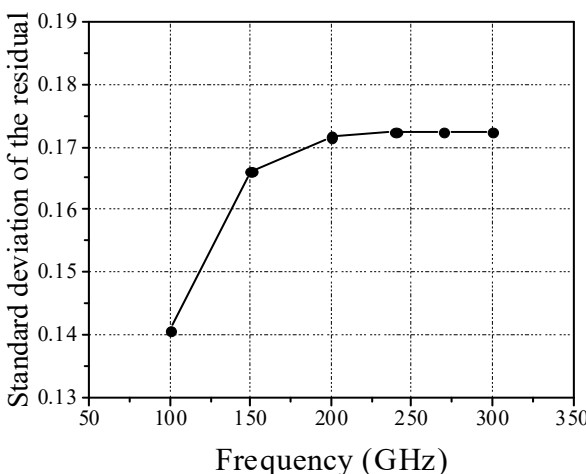

**Figure 4.** Standard deviation of the residual.

**Table 1.** Coefficient values and RMSE based on our DSD model at 240, 270, and 300 GHz.

| Frequency | 240 GHz | 270 GHz | 300 GHz |
|---|---|---|---|
| $k$ | 1.5814 | 1.5640 | 1.5466 |
| $\alpha$ | 0.6620 | 0.6620 | 0.6621 |
| RMSE | 2.8977 | 2.8646 | 2.8331 |

## 3. Empirical Model for Rain Attenuation

The measurement system consists of a transmitter (Tx), a receiver (Rx), and weather measurement units. The transmitting equipment generates the continuous wave signal at 240, 270, and 300 GHz, respectively, and Rx measures the peak levels of the spectrum. The weather measurement unit collects rain intensity data with atmospheric temperature, humidity, and snowfall to distinguish rain attenuation and atmospheric gas attenuation due to dry air and water vapor. The data are sampled every 20 s. The Tx and the weather station were installed on the rooftop of RRA in Korea from March 2021 to February 2022, as shown in Figure 5a. The path length between Tx and Rx is 650 m on the line-of-sight path. Also, Figure 5b shows block diagrams of the transmitter and receiver. Table 2 presents the measurement system parameters of Tx and Rx. The gain and beam width of the Tx and Rx antennas are the same at 46 dBi and 1 degree, respectively. As the path between Tx and Rx is an extremely long distance in view of measurement frequencies, the received power can be measured by the high gain antenna with a beam width of 1 degree. The linear polarization of the Tx and Rx antennas is vertical. For horizontal and vertical polarizations, the specific rain attenuations are approximately the same above 100 GHz in the recommendation ITU-R P.838-3.

Thus, vertically polarized antennas for the transmitter and receiver were utilized in the measurement system. Additionally, if the rainfall rate exceeds 0.25 mm/h, we judge that rainfall has occurred. The reference value of received power is the average received power for 10 min before rain. If the rainfall exceeds 0.25 mm/h within 10 min, it is judged to be a continuous rainfall event. Conversely, if the rainfall rate is 0 for 10 min, the rainfall event is judged to have stopped. When snowfall is detected, the rainfall rate is changed to 0 and excluded from attenuation by rainfall.

The rainfall rate ($R$) is defined as the amount of rainfall falling over a certain period of time. In addition, the time availability of $p$% means rainfall rate $x$ mm/h that the time exceeded corresponds to $p$% of the total measurement period. Figure 6 shows the measured rainfall rate distribution of various integration times in southwestern Korea from March 2021 to February 2022. The rainfall rates at time availabilities of 0.1% and 0.01% are 16.3 mm/h and 57.1 mm/h, respectively. The rain attenuation was measured for one

year, and one year corresponds to 525,600 min. Therefore, it means that it rained more than 57.1 mm/h for 52.56 min in the southwestern province of Korea for one year.

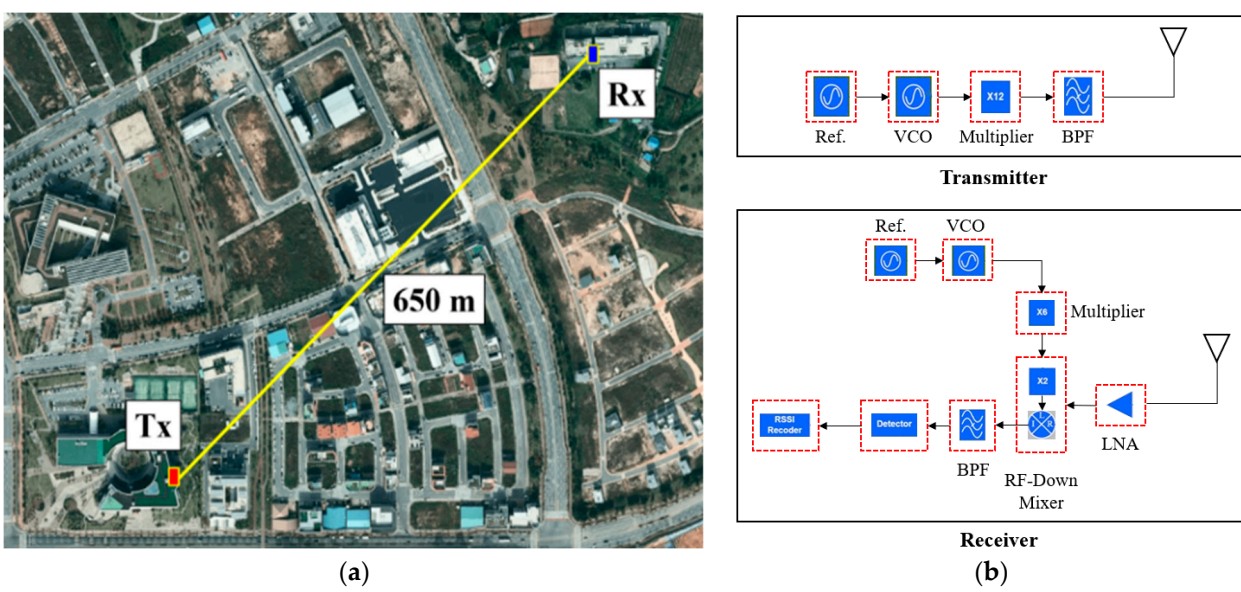

|     | (a) | (b) |

**Figure 5.** (**a**) Terrestrial path link of Tx and Rx (Satellite view); (**b**) block diagrams of transmitter and receiver.

**Table 2.** Measurement system parameters.

| Frequency | | | 240 GHz, 270 GHz and 300 GHz |
| --- | --- | --- | --- |
| **Polarization** | | | **Linear** |
| Antenna | Tx | Type | directional |
| | | Gain | 46 dBi |
| | | Beam width (3 dB) | 1 degree |
| | | Polarization | vertical |
| | Rx | Type | directional |
| | | Gain | 46 dBi |
| | | Beam width (3 dB) | 1 degree |
| | | Polarization | vertical |

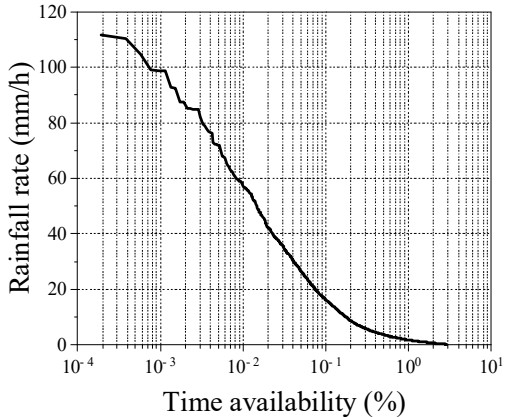

**Figure 6.** Rainfall rate distribution of various integration times.

In order to distinguish rain attenuation and atmospheric gas attenuation, the atmospheric gas attenuation simultaneously should be calculated as shown in reference [29] using temperature, humidity, and air pressure measured at the same time. The specific gases attenuation is given by

$$\gamma = \gamma_o + \gamma_w = 0.1820 f \left( N''_{Oxygen}(f) + N''_{WaterVapor}(f) \right) \tag{8}$$

where $\gamma_o$ and $\gamma_w$ are the specific attenuation (dB/km) due to dry air (oxygen, pressure-induced nitrogen, and non-resonant Debye attenuation) and water vapor, respectively. $f$ is the frequency (GHz). Also, $N''_{Oxygen}(f) + N''_{Water\ Vapor}(f)$ can be calculated by

$$N''_{Oxygen}(f) = \Sigma_{i(Oxygen)} S_i F_i + N''_D(f) \tag{9}$$

$$N''_{Water\ Vapor}(f) = \Sigma_{i(Water\ Vapor)} S_i F_i \tag{10}$$

$S_i$ is the strength of the $i$th oxygen or water vapor line, $F_i$ is the oxygen or water vapor line shape factor, and the summations extend over all the spectral lines in reference [29]. $N''_D(f)$ is the dry continuum due to pressure-induced nitrogen absorption and the Debye spectrum. Figure 7 shows the specific gas attenuation against measurement time per year. As shown in Figure 7, the specific gas attenuation value increases during the rainy season in Korea due to high humidity at all measurement frequencies. After removing the gas attenuation, the specific rain attenuation is obtained from the received power at 240, 270, and 300 GHz, as shown in Figure 8. To compare the specific rain attenuation ($\gamma_R$) between our empirical model and the ITU-R P.838.3 model in Section 4, we have performed a curve fit using regression analysis. It is assumed that the shape of the curve-fitting graph is Equation (7) to match the measured data effectively and to have a similar form to the recommendation ITU-R P.838-3. Table 3 indicates the coefficient values and root mean square errors (RMSEs) based on the empirical model at 240, 270, and 300 GHz. The $k$ and $\alpha$ value of the empirical model is larger and smaller than those of recommendation ITU-R P.838-3, respectively.

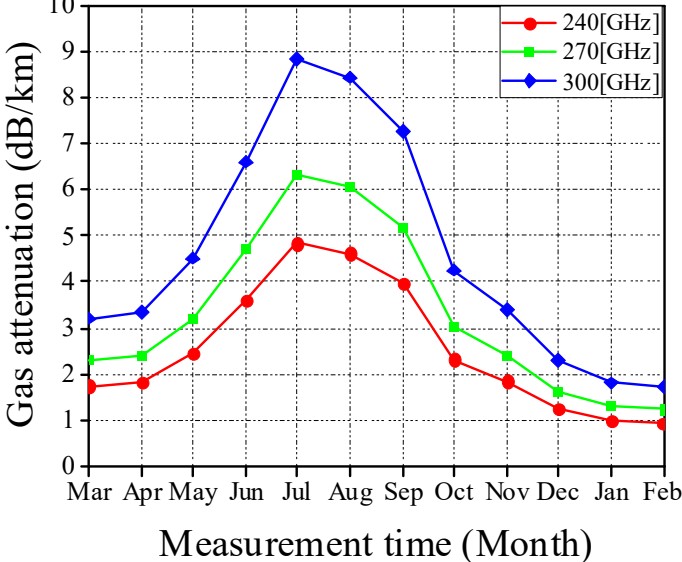

**Figure 7.** Specific gas attenuation against measurement time per year.

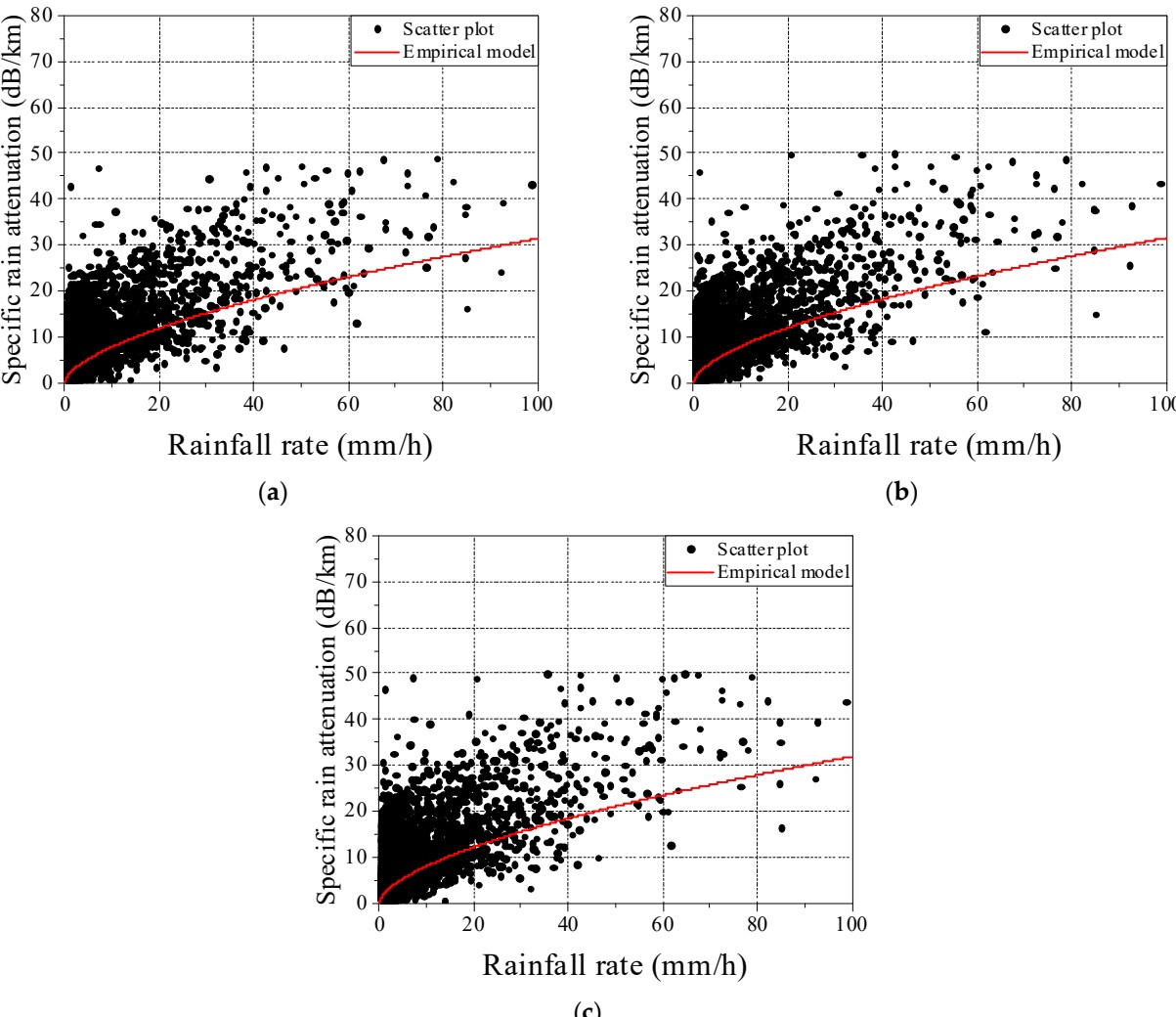

**Figure 8.** Specific rain attenuation based on empirical model: (**a**) 240 GHz, (**b**) 270 GHz, and (**c**) 300 GHz.

**Table 3.** Coefficient values and RMSE based on empirical model at 240, 270, and 300 GHz.

| Frequency | 240 GHz | 270 GHz | 300 GHz |
|:---:|:---:|:---:|:---:|
| $k$ | 1.9463 | 1.9892 | 2.0225 |
| $\alpha$ | 0.6040 | 0.6001 | 0.5989 |
| RMSE | 3.1524 | 3.2279 | 3.4366 |

## 4. Comparison of Rain Attenuation

Figure 9 shows the comparison of specific rain attenuation between the empirical model, measured DSD model, and ITU-R recommendation. The prediction of ITU-R recommendation for vertical polarization is because the Tx/Rx antennas for the empirical model have characteristics of vertical polarization. Generally, the specific rain attenuation for horizontal polarization is larger than that for vertical polarization, but they are almost the same at more than 100 GHz. As shown in Figure 9, the measured DSD model closely matches the ITU-R recommendation up to the rainfall rate of 5 mm/h. On the other hand, the empirical model using the Tx/Rx antenna system is slightly higher compared to the ITU-R recommendation at the $R$ = 5 mm/h. As the rainfall rate increases, the difference between our results and ITU-R recommendation increases. Moreover, the specific rain attenuation of the empirical model is slightly larger than that of the DSD model at a low rainfall rate.

The intersection of rainfall rate increases as the frequency increases. Finally, Table 4 shows the comparison of coefficient values between the empirical model, DSD model, and ITU-R recommendation at 240, 270, and 300 GHz. Also, Table 5 shows the comparison of the characteristics between the previously published research and our research.

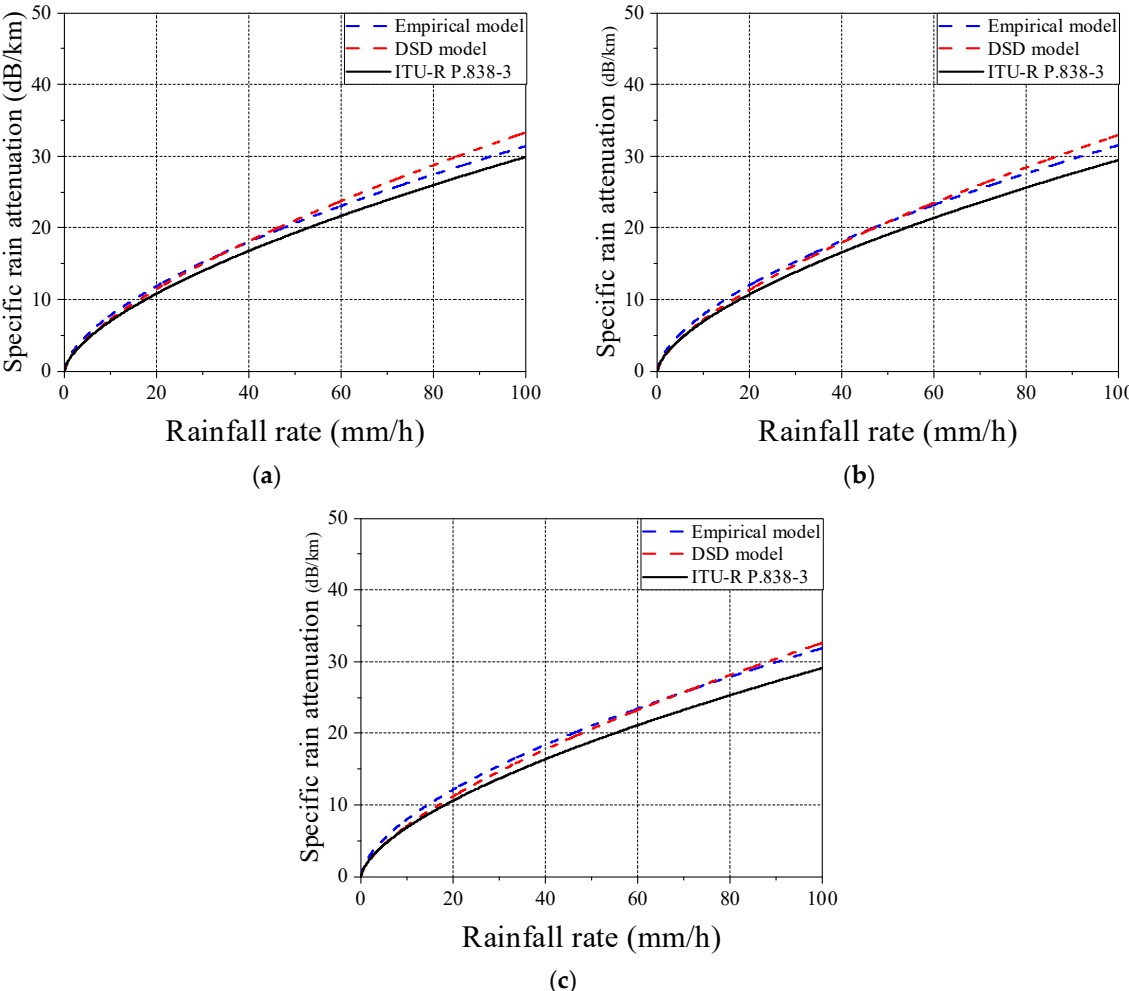

**Figure 9.** Comparison of specific rain attenuation between empirical model measured DSD model and ITU-R recommendation: (**a**) 240 GHz, (**b**) 270 GHz, and (**c**) 300 GHz.

**Table 4.** Comparison of coefficient values between DSD model, empirical, and ITU-R recommendation at 240, 270, and 300 GHz.

| Model | Frequency | $k$ | $\alpha$ |
|---|---|---|---|
| DSD model | 240 GHz | 1.5814 | 0.6620 |
| | 270 GHz | 1.5640 | 0.6620 |
| | 300 GHz | 1.5466 | 0.6621 |
| Empirical model | 240 GHz | 1.9463 | 0.6040 |
| | 270 GHz | 1.9892 | 0.6001 |
| | 300 GHz | 2.0225 | 0.5989 |
| ITU-R recommendation (Vertical/Horizontal) | 240 GHz | 1.6473/1.6434 | 0.6295/0.6337 |
| | 270 GHz | 1.6399/1.6380 | 0.6274/0.6314 |
| | 300 GHz | 1.6286/1.6286 | 0.6262/0.6296 |

**Table 5.** Comparison of the characteristics between the previously published research and our research.

| | DSD Data | Comparison with Measured Attenuation | Frequency Band |
| --- | --- | --- | --- |
| Ref. [12] | Measurement | No | 1, 100, and 1000 GHz |
| Ref. [15] | Analytic model | No | 300~1000 GHz |
| Ref. [19] | Analytic model | Yes | 90, 140, 225, 313, and 355 GHz |
| Ref. [21] | Measurement | Yes (Not at same time) | 77 and 300 GHz |
| This work | Measurement | Yes | 240, 270, and 300 GHz |

**5. Conclusions**

In order to verify the expectations of recommendation ITU-R P.838-3 at the low THz frequency band, the specific rain attenuation based on the DSD model is calculated and confirmed by the empirical model at 240, 270, and 300 GHz in this paper. The ECS for specific rain attenuation based on the DSD model is calculated by Mie scattering due to its low THz frequencies. Also, to obtain only specific rain attenuation based on the empirical model, the atmospheric gas attenuation is removed using the weather measurement unit. The received power of the antenna system for the empirical model and DSD of rain have been measured in southwestern Korea for one year between March 2021 and February 2022. From the calculated results, we know that the specific rain attenuation of the empirical model is slightly larger than that of the DSD model at a low rainfall rate. Both specific attenuations for rain are a little larger than that of the ITU-R recommendation. As the rainfall rate increases, the difference between our results and ITU-R recommendation increases. This conclusion would be useful for predicting rain attenuation for terrestrial wireless links operating at 240, 270, and 300 GHz.

**Author Contributions:** Conceptualization, J.K. and S.P.; methodology, J.L.; software, J.L.; validation, J.K., S.P. and J.L.; formal analysis, Y.K.; investigation, J.L.; writing—original draft preparation, J.L.; writing—review and editing, J.L.; visualization, J.L.; supervision, J.K., J.O. and Y.Y. All authors have read and agreed to the published version of the manuscript.

**Funding:** This work was supported by the Institute for Information & Communications Technology Planning & Evaluation (IITP) grant funded by the Korea Government (MSIT) (No. 2021-0-00335, Development of close proximity multipath propagation model for 275~450 GHz band). The authors would like to thank Hee Jun Park, with the National Radio Research Agency (RRA), Naju-Si Jeollannam-do, Korea, for measurement data and discussion.

**Data Availability Statement:** Data are contained within the article.

**Conflicts of Interest:** The authors declare no conflict of interest.

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
