# Peer review of "Rain Attenuations Based on Drop Size Distribution (DSD) Model and Empirical Model at Low THz Frequencies"

_electronics, doi:10.3390/electronics13010009_

Round 1

Reviewer 1 Report

Comments and Suggestions for Authors

1)      The work of the manuscript is meaningful. However, my primary concern centers around the novelty of the method developed in the manuscript.

2)      Perhaps due to the error of the measurement system and the selected terahertz band, some data are too consistent and do not show significant differences. For example, (a) and (b) in Figure 2 are almost identical. If the two images are superimposed, the approximate effect is as shown below. Almost no difference can be seen. It is suggested that the author improve the measurement system and data acquisition system.

Reviewer 2 Report

Comments and Suggestions for Authors

This paper exhibits an effort to derive a ran attenuation framework to validate the expectations outlined in ITU-R P.838-3 for recommendations. Given the specific goal, this work may exhibit itself as a possibly workable experimental model; however, it lacks detailed analyses to explain why their key conclusions would be possible: 1) the specific rain attenuation of empirical model is slightly larger than that of DSD model at low rainfall rate, and 2) both specific attenuations for rain are a little larger than that of the ITU-R recommendation. Moreover, as indicated by the authors as well, there are many related works on attenuation for rain in the low-THz band being done in recent years, as summarized from [12] to [23]. However, no further introductions to these related works are given, and no comparisons to these works are given in the numerical results.

Accordingly, to enhance the credibility and potential contributions of this work, the following suggestions for publication are suggested:

1. Comprehensive Introduction to Related Works: Provide a thorough introduction to all related works, outlining their methodologies, results, and highlighting the strengths and weaknesses in comparison to the proposed framework.

2. Numerical Comparison with State-of-the-Art Works: Include a comparison with at least one state-of-the-art work, presenting numerical results that showcase the potential advantages of the proposed methods over existing approaches.

Comments on the Quality of English Language

Minor corrections to the writing are required.

Reviewer 3 Report

Comments and Suggestions for Authors

The manuscript, titled 'Rain Attenuations Based on Drop Size Distribution (DSD) Model and Empirical Model at Low-THz Frequencies,' proposes a study on rain attenuation for various rainfall rates at low-THz frequencies. The study is well-motivated, and the methodology is reasonably well-explained. I have only minor comments that should be addressed before publishing this work.

11)     How did the authors handle synchronization issues between the TX and RX signals?

22)     It would be helpful if the authors could include a table comparing their work with others.

33)     Could the authors include further discussion regarding the RF circuits used on the receiver side and the transmitter side? A block diagram and/or photos of both chains would suffice.

Reviewer 4 Report

Comments and Suggestions for Authors

The performance of high GHz wireless links in rain is an important topic.

The paper has a good quality data, and sound conclusions, thus it rates as having high scientific merit, however the authors need to improve the data presentation and English language presentation.

Abstract

Line there is no explanation provided for the origin of recommendation ITU-R P.838-3, or the acronym RMSEs

The English in lines 19 to 24 is awkward

Introduction

Line 55 there is no reference provided for recommendation ITU-R P.838-3. Until line 71.

More explanation of the detail of ITU-R P.838-3 is needed. 

Section 2. “Drop size distribution (DSD) model for rain attenuation” is laid out in a very difficult manner. It is not an important section and could be significantly reduced an the data analysis put in sect 3.

The form of the gamma, Weibull, and MP DSD models is not obvious without resorting to references.

The reference to the fact that the gamma MP model are a the best fit should be accompanied by a statistical analysis.

Presumably, ND iof the Y axis stands for “number of drops”, in Fig 1, but this is not explicitly stated.

It is not clear in Fig. 2 how the 240, 270, 300 GHz are represented the caption does not match the text.

It is not clear in Fig. 2 how the scatter plot was obtained. It seems to be different to the one in Section 3.

It is not clear in Fig. 2 how gamma, Weibull, and MP DSD models are represented

It is not clear in Fig. 2 how (a) 5 mm/h (b) 10 mm/h (c) 15 mm/h (d) 20 mm/h. are represented

Line 92 confusingly, Gamma [27] is in capitals, suggesting it is the name of the author.

Line 103 “S(0) is forward scattering coefficient and k is wavenumber” needs “the” inserted

Line 111 and 112 “Bessel function of first kind” needs “the” inserted. 

Section 3. “Empirical model for rain attenuation” is a much better set out section when compared to Section 2.

The English in this section is awkward:  “Thus, … Moreover…it is determined that rainfall occurs… On the contrary…specific gases attenuation, etc” do not flow well.

The logarithmic x-axis in Fig. 6. “Rainfall rate versus time availability” caption does not convey information well.

If the scatter plot in Fig. 8 is of a year’s readings at 20 second intervals it represents a total of 1.5 million data points, in which case it is not a suitable graphic, since the lower left of the graph becomes saturated very quicky.

Section 4. Comparison of rain attenuation and Section 5. Conclusion

Apart from minor English language flaws, theses sections are satisfactory.

Comments on the Quality of English Language

See above, needs an English scientific editor. 

Round 2

Reviewer 1 Report

Comments and Suggestions for Authors

The authors have responded to my concerns.

Author Response

We again appreciate your time and consideration in anticipation.

Reviewer 2 Report

Comments and Suggestions for Authors

No further comments.

Comments on the Quality of English Language

No further comments.

Author Response

Some improvements will be made when submitting the final version. We again appreciate your time and consideration in anticipation.

Reviewer 4 Report

Comments and Suggestions for Authors

Much improved, happy to publish.

Author Response

(The authors gave the same response as above.)
